# LDL Receptor-Related Protein 1B Polymorphisms Associated with Increased Risk of Lymph Node Metastasis in Oral Cancer Group with Diabetes Mellitus

**DOI:** 10.3390/ijms25073963

**Published:** 2024-04-02

**Authors:** Liang-Cheng Chen, Yu-Sheng Lo, Hsin-Yu Ho, Chia-Chieh Lin, Yi-Ching Chuang, Wei-Chen Chang, Ming-Ju Hsieh

**Affiliations:** 1Division of Oral & Maxillofacial Surgery, Dental Department, Changhua Christian Hospital, Changhua 500, Taiwan; 2Oral Cancer Research Center, Changhua Christian Hospital, Changhua 500, Taiwan; 3Doctoral Program in Tissue Engineering and Regenerative Medicine, College of Medicine, National Chung Hsing University, Taichung 402, Taiwan; 4Graduate Institute of Biomedical Sciences, China Medical University, Taichung 404, Taiwan

**Keywords:** LRP1B, OSCC, diabetes mellitus

## Abstract

Oral cancer ranks fourth among malignancies among Taiwanese men and is the eighth most common cancer among men worldwide in terms of general diagnosis. The purpose of the current study was to investigate how low-density lipoprotein receptor-related protein 1B (LDL receptor related protein 1B; LRP1B) gene polymorphisms affect oral squamous cell carcinoma (OSCC) risk and progression in individuals with diabetes mellitus (DM). Three LRP1B single-nucleotide polymorphisms (SNPs), including rs10496915, rs431809, and rs6742944, were evaluated in 311 OSCC cases and 300 controls. Between the case and control groups, we found no evidence of a significant correlation between the risk of OSCC and any of the three specific SNPs. Nevertheless, in evaluating the clinicopathological criteria, individuals with DM who possess a minimum of one minor allele of rs10496915 (AC + CC; *p* = 0.046) were significantly associated with tumor size compared with those with homozygous major alleles (AA). Similarly, compared to genotypes homologous for the main allele (GG), rs6742944 genotypes (GA + AA; *p* = 0.010) were more likely to develop lymph node metastases. The tongue and the rs6742944 genotypes (GA + AA) exhibited higher rates of advanced clinical stages (*p* = 0.024) and lymph node metastases (*p* = 0.007) when compared to homozygous alleles (GG). LRP1B genetic polymorphisms appear to be prognostic and diagnostic markers for OSCC and DM, as well as contributing to genetic profiling research for personalized medicine.

## 1. Introduction

Oral squamous cell carcinoma (OSCC) is a major oral cancer subtype and a leading cause of death in Taiwan. It accounts for about 95% of oral cancer cases in Taiwan [1]. A number of risk factors, including alcohol intake, betel quid use, and tobacco use, contribute to the development of oral cancer [2]. Systematic analysis of candidate gene associated studies have suggested that single-nucleotide polymorphisms (SNPs) in genes involved in DNA repair, carcinogen metabolism, cell cycle control, extracellular matrix alteration, and folate metabolism might be linked with increased oral cancer susceptibility [3,4]. Over the past ten years, there has been a significant improvement in the chances of better OSCC diagnosis and therapy; however, a thorough understanding of the pathophysiology of OSCC remains a significant obstacle.

In human cancer, low-density lipoprotein receptor-related protein 1B (LDL receptor related protein 1B; LRP1B) is one of the most frequently mutated genes. Most often, it has been considered a potential tumor suppressor due to its frequent inactivation caused by several genetic and epigenetic processes [5]. LRP is highly expressed in a number of organs and is involved in the pathophysiology of Alzheimer’s disease, blood coagulation, cell adhesion and migration, and lipoprotein catabolism [6,7,8]. Changes in the tumor environment are caused by epigenetic control of LRP1B. Examples of these include the abnormal methylation of LRP1B in gastric cancer, acute lymphoblastic leukemia, and lung adenocarcinoma [9,10,11,12]. LRP1B downregulation in colon cancer tissues prevents colon cancer cells from proliferating, migrating, and metastasizing [13]. LRP1B somatic alterations were found in more than 20% of tumor types identified in the Cancer Genome Atlas data, including non-small-cell lung cancer, melanoma, esophageal, gastric, head and neck cancers, and bladder cancers [5,14,15]. Previous studies have shown that there are extensive host genotype–microbe interactions in the oral cavity, with LRP1B rs10496915 having significance with oral dialister abundance implicated in lipoprotein metabolism [16]. Furthermore, participants carrying only the LRP1B rs80306347 allele had an increased risk of progressing to Parkinson’s disease dementia compared with non-carriers [17]. In particular, the LRP1B gene has been associated with obesity in genetic association studies, with a bimodal pattern of CpG methylation and dependence on genotype rs431809 [18].

Recently, research has also looked into the possible link between diabetes mellitus (DM) and oral cancer. It was shown that individuals with DM had a higher chance of developing precancerous lesions and mouth cancer [19]. It has been observed that the treatment outcomes of cancers are impacted by DM, a common chronic metabolic illness [20]. OSCC patients with DM had lower overall survival, recurrence-free survival, and cancer-specific survival rates compared with non-diabetics (adjusted hazard ratio [HR] = 2.22, 2.42, and 2.16, respectively) even in less-aggressive tumor stages (stage I and II) [20]. Combining these findings from multiple studies indicates that LRP1B plays suppressive functions in the development of cancer and that regaining LRP1B function would be a viable approach to treating the disease. However, the relation between LRP1B and OSCC is still poorly understood. In this present study we investigated the genetic polymorphisms of LRP1B in OSCC patients with DM by SNP genotyping analysis.

## 2. Results

### 2.1. Subject Characteristics

Table 1 displays the specifics of the OSCC characters. In order to examine the potential correlation between LRP1B gene polymorphisms and the onset of oral carcinogenesis, 311 patients with OSCC and 300 cancer-free controls were enrolled in this case–control study. Between the case and control groups, there were notable variations in the frequency of alcohol drinking, betel quid chewing, and cigarette smoking. It was subsequently found that the frequency of DM differed significantly between cases and controls, with DM observed in 29.5% of the OSCC cohort. OSCC patients will be classified into tongue cancer and buccal cancer to understand their distribution in clinical characterization. Lymph node metastasis in patients with tongue cancer and buccal cancer was 37.9% and 25.3%, respectively, while the proportions of distant metastasis were 11.2% and 4.7%.

### 2.2. Association of LRP1B SNP with the Progression of Oral Cancer

To test the possible association of LRP1B gene polymorphisms with the development of OSCC, three SNPs (rs10496915, rs431809, and rs6742944) were genotyped in this study. We looked at the genotype frequency distribution for each SNP in OSCC patients and cancer-free controls. Between the case and control groups, there was no discernible relationship found between these LRP1B polymorphisms and the likelihood of developing oral cancer (Table 2).

The clinical status and LRP1B genotype frequency of the OSCC group were also discussed, for which three SNPs of LRP1B were genotyped (rs10496915, rs431809, and rs6742944). Between the case and control groups, there was no discernible correlation between these SNPs and the risk of oral cancer (Table 3).

As part of our study, we also investigated whether LRP1B gene polymorphisms have a common impact on clinicopathological characteristics in patients with OSCC diagnosed with DM. For patients who had DM (*n* = 85), a significant association of rs10496915 genotypes (AC  +  CC; *p* = 0.046) with larger tumor size in comparison with homozygotes for the major allele (AA). Comparing genotypes homologous for the main allele (GG) with those of rs6742944 (GA + AA; *p* = 0.010), the latter group was more likely to experience lymph node metastases (Table 4). The combination of DM and LRP1B gene variants in OSCC patients may have an impact on the disease’s course, according to our data.

As for the distribution of tumor sites, samples from the tongue and the rs6742944 (GA + AA) genotypes exhibited higher risk of advanced clinical stage and progression of lymph node metastasis compared with those with the homozygotes allele (GG) (Table 5). No such results were found in buccal cancer.

## 3. Discussion

LRP1B is a member of the LDL receptor family. The downregulation of LRP1B was observed in non-small-cell lung cancer cell lines [21] and in renal cell cancer tissues and cell lines [22]. Wang and colleagues have found that LRP1B contributed to 12.3% of hepatocellular carcinoma patients with mutated genes in the Chinese cohort [23]. In this study, we selected three LRP1B gene polymorphisms (rs10496915, rs431809, and rs6742944) to compare their allelic distributions among cancer-free subjects, patients with OSCC, and OSCC patients with DM. Alcohol consumption, betel quid chewing, and cigarette smoking are considered as the main risk factors associated with development of OSCC [24]. In our study, statistically significant associations of these risk factors were found in 311 oral cancer patients compared with the controls, respectively (*p* < 0.0001, Table 1). Diabetes patients have an increased risk of precancerous lesions and oral cancer. But the association between diabetes and head and neck cancers is still controversial [19,25]. Tseng et al. also found a higher risk of head and neck cancer in the DM cohort than in the non-DM cohort (HR, 1.16; 95% CI, 1.11–1.22) [26]. In this study, DM was significantly associated with oral cancer patients compared with the control cohort (*p* < 0.0001, Table 1).

We also looked at the relationship between oral cancer susceptibility and LRP1B genotypic frequencies. This study indicates that LRP1B polymorphisms have an elevated risk of mouth cancer but a restricted carcinogenic effect because no significant correlations were seen between the controls and the oral cancer patients (Table 2). Interestingly, after we analyzed the LRP1B genotypic frequencies among OSCC patients in our study, no statistically significant association was found between the oral cancer patients and the normal controls of all three LRP1B gene polymorphisms (rs10496915, rs431809, and rs6742944) (Table 3). We found that amongst OSCC patients who had DM, the AA allele at rs10496915 was significantly associated with tumor size, and the GG allele at rs6742944 was significantly associated with lymph node metastasis (Table 4).

Oral cancer, which occurs in the mouth, lip and tongue, causes significant morbidity and mortality [27]. DM patients had considerably greater probabilities of having nasopharyngeal carcinoma, oropharyngeal cancer, and oral cavity cancer [25]. We further analyzed the risk association of LRP1B rs6742944 for tongue and buccal mucosa cancers among DM cohorts with oral cancer. Samples from the tongue and the rs6742944 (GA + AA) genotypes showed higher risk of advanced clinical stage and progression of lymph node metastasis.

Due to the lack of information on the underlying disease process in the Taiwanese database, the sample in this study for oral cancer–diabetes associations was inadequate to study disease pathways. Additionally, the recall and willingness of patients play a major role in long-term survival. Further studies should be conducted using larger sample sizes and longer follow-ups to examine whether LRP1B SNPs are associated with OSCC in the future.

## 4. Materials and Methods

### 4.1. Patients and Samples

Samples for this study were collected at Changhua Christian Hospital. Regulatory approval for this study was obtained from the Institutional Review Board (IRB) of Changhua Christian Hospital under the number 130616. The research group included 311 patients diagnosed with OSCC and 300 cancer-free patients in the control group at Changhua Christian Hospital from 2014 to 2023. In this study, a total of 611 cases were collected, and all the patients who participated signed a written informed consent form before starting the project. We obtained statistical data on age and personal habits (including betel nuts, smoking, and alcohol consumption) from medical documents. Additionally, AJCC No. 8 is also used to discuss the judgment of clinical stage, tumor/lymph node/metastasis stage, and degree of cell differentiation [28]. For the LRP1B polymorphisms, the investigator collected venous blood samples and stored them in tubes containing K3-ethylene diamine tetraacetic acid (EDTA). The blood samples are then cryogenically centrifuged and stored in a −80 °C laboratory freezer for analysis.

### 4.2. DNA Extraction and Analysis LRP1B SNP with Real-Time PCR

This study demonstrates that these LRP1B SNPs use ABI SNP browsers to select appropriate sites and exclude linkage disequilibrium (LD, linkage disequilibrium) sites through the LD link website. By using the National Institutes of Health Variation Viewer, the minor allele frequencies (MAF) with fewer genetic loci were excluded. A selection of options with an LD-LINK score of more than 0.8 and a minimum MAF of 10% was eliminated. The three LRP1B SNPs rs10496915 (A/C), rs431809 (G/T), and rs6742944 (G/A) obtained through the above analysis were included in the analysis model, and these studies have shown that they increase the risk of various diseases [16,18,29]. Each of these genotyping assays was ordered from Applied Biosystems with a TaqMan-minor groove binder (MGB) moiety genotyping assay mix. The probe IDs for TaqMan-SNP Genotyping Assay Data Sheets were C_29842934 (rs10496915), C_822593 (rs431809), and C_2115669 (rs6742944), and all probes were stored at −20 °C. In each TaqMan-MGB genotyping mix, one primer matched perfectly to the wild-type sequence variant labeled with VIC, while the second primer matched to the mutant (SNP) sequence variant labeled with 6-carboxyfluorescein (FAM) [30]. Similarly, in our previous research, we used DNA extraction, preservation, and analysis techniques [31,32]. EDTA-containing sterile tubes containing whole blood samples were collected from patients and immediately centrifuged and stored at −80 °C. The genomic DNA was extracted from peripheral blood leukocytes using a QIAamp DNA blood mini kit according to the manufacturer’s protocol (Qiagen, Valencia, CA, USA), and then dissolved in TE buffer and stored at −20 °C. Optical density was quantified based on 260 nm wavelength measurements. The three polymorphisms rs10496915 (A/C), rs431809 (G/T), and rs6742944 (G/A) of the potential of the LRP1B gene were determined by quantitative real-time PCR using the ABI StepOne real-time PCR system (Applied Biosystems, Foster City, CA, USA) and analyzed using StepOne Software v2.3. To create each reaction, 2.5 µL of TaqMan Genotyping Master Mix (Thermo Fisher Scientific Inc., Waltham, MA, USA), 0.125 µL of TaqMan probe mix, and 30 ng genomic DNA were combined in a final volume of 5 µL. In the real-time PCR procedure, the first step was denaturation at 95 °C for 10 min, followed by 40 amplification cycles at 95 °C for 15 s and 60 °C for 1 min.

### 4.3. Statistical Analysis

We used IBM SPSS Statistics v22.0 (IBM, Armonk, NY, USA) to perform analyses in our study similar to previous papers [32]. First, the demographic and laboratory data between the non-OSCC group and the OSCC group were shown using descriptive analysis including mean, standard deviation (SD), and percentage, and evaluated using the exact Mann–Whitney U test difference between the two groups. Logistic regression models were then used to analyze the odds ratios (OR) and the associated 95% confidence intervals (CI) of the LRP1B SNP polymorphism distribution between non-OSCC and OSCC populations. Additionally, multiple logistic regression models were used to calculate the adjusted odds ratio (AOR) between the two groups after chewing betel nuts, alcohol, tobacco consumption, and DM. Following this analysis, we divided OSCC patients with DM into tongue cancer and buccal cancer, and analyzed the correlation between LRP1B SNP rs6742944 and clinicopathological characteristics of OSCC to generate ORs with 95% CI.

## 5. Conclusions

This is the first study to show how DM and LRP1B gene polymorphisms interact to influence the development of oral cancer. Although the future utility of rs6742944 as a confirmatory factor is unknown, there is a possibility that our study of LRP1B polymorphisms may provide new insight into developing these markers as useful prognostic markers for the treatment of OSCC in the future.

## Figures and Tables

**Table 1 ijms-25-03963-t001:** The distributions of demographical characteristics and clinical parameters in 300 controls and 311 cases with OSCC.

Variable	Control (N = 300)	Patients (N = 311)		*p* Value
**Age (yrs.)**	53.92 ± 7.77	53.50 ± 10.26		
<54	148 (49.3%)	156 (50.2%)		*p* = 0.669
≥54	152 (50.7%)	155 (49.8%)		
**Betel nut chewing**				
No	289 (96.3%)	95 (30.5%)		*p* < 0.0001 *
Yes	11 (3.7%)	216 (69.5%)		
**Cigarette smoking**				
No	277 (92.3%)	48 (15.4%)		*p* < 0.0001 *
Yes	23 (7.7%)	263 (84.6%)		
**Alcohol drinking**				
No	292 (97.3%)	183 (58.8%)		*p* < 0.0001 *
Yes	8 (2.7%)	128 (41.2%)		
**Diabetes mellitus**				
No	246 (87.5%)	203 (70.5%)		*p* < 0.0001 *
Yes	35 (12.5%)	85 (29.5%)		
**Stage**		**Tongue**	**Buccal**	
I + II		79 (49.1%)	80 (53.3%)	
III + IV		82 (50.9%)	70 (46.7%)	
**Tumor T status**				
T1 + T2		101 (62.7%)	98 (65.3%)	
T3 + T4		60 (37.3%)	52 (34.7%)	
**Lymph node status**				
N0		100 (62.1%)	112 (74.7%)	
N1 + N2 + N3		61 (37.9%)	38 (25.3%)	
**Metastasis**				
M0		143 (88.8%)	143 (95.3%)	
M1		18 (11.2%)	7 (4.7%)	
**Cell differentiation**				
Well differentiated		17 (10.6%)	32 (21.3%)	
Moderately or poorly differentiated		144 (89.4%)	118 (78.7%)	

N: number. * *p* value < 0.05 as statistically significant.

**Table 2 ijms-25-03963-t002:** The distribution of genotype frequencies in LRP1B SNPs in cases of the OSCC group.

Variable	Control (N = 300)	Patients (N = 311)	OR ^a^ (95% CI)	AOR ^b^ (95% CI)	AOR ^c^ (95% CI)
**rs10496915**					
AA	209 (69.7%)	228 (73.3%)	1.000	1.000	1.000
AC	82 (27.3%)	75 (24.1%)	1.193 (0.828–1.718)	1.119 (0.613–2.041)	1.239 (0.841–1.826)
CC	9 (3.0%)	8 (2.6%)	1.227 (0.465–3.240)	2.764 (0.554–13.799)	1.104 (0.360–3.382)
AC + CC	91 (30.3%)	83 (26.7%)	1.196 (0.841–1.700)	1.220 (0.682–2.183)	1.227 (0.843–1.786)
**rs431809**					
GG	198 (66.0%)	189 (60.8%)	1.000	1.000	1.000
GT	91 (30.3%)	107 (34.4%)	0.812 (0.576–1.144)	1.067 (0.599–1.900)	0.842 (0.585–1.212)
TT	11 (3.7%)	15 (4.8%)	0.700 (0.314–1.563)	0.758 (0.201–2.852)	0.555 (0.233–1.322)
GT + TT	102 (34.0%)	122 (39.2%)	0.798 (0.574–1.110)	1.025 (0.590–1.781)	0.803 (0.565–1.139)
**rs6742944**					
GG	212 (70.7%)	238 (76.5%)	1.000	1.000	1.000
GA	83 (27.7%)	69 (22.2%)	1.350 (0.934–1.953)	1.166 (0.627–2.169)	1.263 (0.858–1.859)
AA	5 (1.7%)	4 (1.3%)	1.403 (0.372–5.294)	3.850 (0.449–33.016)	1.310 (0.308–5.572)
GA + AA	88 (29.3%)	73 (23.5%)	1.353 (0.943–1.942)	1.251 (0.681–2.298)	1.265 (0.866–1.848)

N: number. The rs10496915, rs431809, and rs6742944 of major/minor alleles are A/C, G/T, and G/A, respectively. ^a^ The odds ratio (OR) with their 95% confidence intervals were estimated by logistic regression models. ^b^ The adjusted odds ratio (AOR) with their 95% confidence intervals were estimated by multiple logistic regression models after controlling for betel nut chewing, alcohol consumption, and tobacco consumption. ^c^ The adjusted odds ratio (AOR) with their 95% confidence intervals were estimated by multiple logistic regression models after controlling for DM.

**Table 3 ijms-25-03963-t003:** Clinical statuses and LRP1B genotype frequencies in cases of the OSCC group.

Variable	LRP1B								
	rs10496915 (N = 311)			rs431809 (N = 311)			rs6742944 (N = 311)		
	AA (%) (N = 228)	AC + CC (%) (N = 83)	*p* Value	GG (%) (N = 189)	GT + TT (%) (N = 122)	*p* Value	GG (%) (N = 238)	GA + AA (%) (N = 73)	*p* Value
**Clinical stage**									
Stage I/II	116 (50.9%)	43 (51.8%)	*p* = 0.885	101 (53.4%)	58 (47.5%)	*p* = 0.310	126 (52.9%)	33 (45.2%)	*p* = 0.248
Stage III/IV	112 (49.1%)	40 (48.2%)		88 (46.6%)	64 (52.5%)		112 (47.1%)	40 (54.8%)	
**Tumor size**									
T1 + T2	144 (63.2%)	55 (66.3%)	*p* = 0.614	125 (66.1%)	74 (60.7%)	*p* = 0.326	153 (64.3%)	46 (63.0%)	*p* = 0.843
T3 + T4	84 (36.8%)	28 (33.7%)		64 (33.9%)	48 (39.3%)		85 (35.7%)	27 (37.0%)	
**Lymph node metastasis**									
No	156 (68.4%)	56 (67.5%)	*p* = 0.873	130 (68.8%)	82 (67.2%)	*p* = 0.772	166 (69.7%)	46 (63.0%)	*p* = 0.281
Yes	72 (31.6%)	27 (32.5%)		59 (31.2%)	40 (32.8%)		72 (30.3%)	27 (37.0%)	
**Distant metastasis**									
No	212 (93.0%)	74 (89.2%)	*p* = 0.276	174 (92.1%)	112 (91.8%)	*p* = 0.934	218 (91.6%)	68 (93.2%)	*p* = 0.670
Yes	16 (7.0%)	9 (10.8%)		15 (7.9%)	10 (8.2%)		20 (8.4%)	5 (6.8%)	
**Cell differentiation**									
Well	37 (16.2%)	12 (14.5%)	*p* = 0.705	28 (14.8%)	21 (17.2%)	*p* = 0.571	37 (15.5%)	12 (16.4%)	*p* = 0.855
Moderate/poor	191 (83.8%)	71 (85.5%)		161 (85.2%)	101 (82.8%)		201 (84.5%)	61 (83.6%)	

N: number. The rs10496915, rs431809, and rs6742944 of major/minor alleles are A/C, G/T, and G/A, respectively.

**Table 4 ijms-25-03963-t004:** Clinical status and LRP1B genotype frequencies in cases of the OSCC group among DM.

Variable	LRP1B								
	rs10496915 (N = 85)			rs431809 (N = 85)			rs6742944 (N = 85)		
	AA (%) (N = 63)	AC + CC (%) (N = 22)	*p* Value	GG (%) (N = 49)	GT + TT (%) (N = 36)	*p* Value	GG (%) (N = 63)	GA + AA (%) (N = 22)	*p* Value
**Clinical stage**									
Stage I/II	31 (49.2%)	12 (54.5%)	*p* = 0.667	27 (55.1%)	16 (44.4%)	*p* = 0.332	35 (55.6%)	8 (36.4%)	*p* = 0.125
Stage III/IV	32 (50.8%)	10 (45.5%)		22 (44.9%)	20 (55.6%)		28 (44.4%)	14 (63.6%)	
**Tumor size**									
T1 + T2	36 (57.1%)	18 (81.8%)	*p* = 0.046 *^,a^	33 (67.3%)	21 (58.3%)	*p* = 0.395	39 (61.9%)	15 (68.2%)	*p* = 0.758
T3 + T4	27 (42.9%)	4 (18.2%)		16 (32.7%)	15 (41.7%)		24 (38.1%)	7 (31.8%)	
**Lymph node metastasis**									
No	44 (69.8%)	14 (63.6%)	*p* = 0.591	35 (71.4%)	23 (63.9%)	*p* = 0.461	48 (76.2%)	10 (45.5%)	*p* = 0.010 *^,b^
Yes	19 (30.2%)	8 (36.4%)		14 (28.6%)	13 (36.1%)		15 (23.8%)	12 (54.5%)	
**Distant metastasis**									
No	59 (93.7%)	20 (90.9%)	*p* = 0.667	47 (95.9%)	32 (88.9%)	*p* = 0.229	59 (93.7%)	20 (90.9%)	*p* = 0.667
Yes	4 (6.3%)	2 (9.1%)		2 (4.1%)	4 (11.1%)		4 (6.3%)	2 (9.1%)	
**Cell differentiation**									
Well	13 (20.6%)	2 (9.1%)	*p* = 0.235	7 (14.3%)	8 (22.2%)	*p* = 0.346	12 (19.0%)	3 (13.6%)	*p* = 0.568
Moderate/poor	50 (79.4%)	20 (90.9%)		42 (85.7%)	28 (77.8%)		51 (81.0%)	19 (86.4%)	

N: number. The rs10496915, rs431809, and rs6742944 of major/minor alleles are A/C, G/T, and G/A, respectively. * *p* value < 0.05 as statistically significant. ^a^ OR (95% CI): 0.296 (0.090–0.977); ^b^ OR (95% CI): 3.840 (1.385–10.649).

**Table 5 ijms-25-03963-t005:** OSCC distribution in type 2 DM patients and correlation with LRP1B rs6742944 genotype and clinical status.

Variable	LRP1B (rs6742944)					
	with Tongue (N = 43)			with Buccal (N = 42)		
	GG (%) (N = 34)	GA + AA (%) (N = 9)	*p* Value	GG (%) (N = 29)	GA + AA (%) (N = 13)	*p* Value
**Clinical stage**						
Stage I/II	23 (67.6%)	2 (22.2%)	*p* = 0.024 *^,a^	12 (41.4%)	6 (46.2%)	*p* = 0.773
Stage III/IV	11 (32.4%)	7 (77.8%)		17 (58.6%)	7 (53.8%)	
**Tumor size**						
T1 + T2	25 (73.5%)	5 (55.6%)	*p* = 0.303	14 (48.3%)	10 (76.9%)	*p* = 0.092
T3 + T4	9 (26.5%)	4 (44.4%)		15 (51.7%)	3 (23.1%)	
**Lymph node metastasis**						
No	26 (76.5%)	2 (22.2%)	*p* = 0.007 *^,b^	22 (75.9%)	8 (61.5%)	*p* = 0.346
Yes	8 (23.5%)	7 (77.8%)		7 (24.1%)	5 (38.5%)	
**Distant metastasis**						
No	33 (97.1%)	8 (88.9%)	*p* = 0.334	26 (89.7%)	12 (92.3%)	*p* = 0.787
Yes	1 (2.9%)	1 (11.1%)		3 (10.3%)	1 (7.7%)	
**Cell differentiation**						
Well	5 (14.7%)	0 (0.0%)	*p* = 0.999	7 (24.1%)	3 (23.1%)	*p* = 0.941
Moderate/poor	29 (85.3%)	9 (100.0%)		22 (75.9%)	10 (76.9%)	

N: number. The rs6742944 of major/minor alleles is G/A. * *p* value < 0.05 as statistically significant. ^a^ OR (95% CI): 7.318 (1.300–41.194); ^b^ OR (95% CI): 11.375 (1.957–66.113).

## Data Availability

The datasets generated for this study are available upon request of the corresponding authors.

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
