# Peer review of "LDL Receptor-Related Protein 1B Polymorphisms Associated with Increased Risk of Lymph Node Metastasis in Oral Cancer Group with Diabetes Mellitus"

_ijms, 2024, doi:10.3390/ijms25073963_

Round 1

Reviewer 1 Report

Comments and Suggestions for Authors

This study investigated the possible role of LDL receptor-related protein 1B polymorphisms in the development and prognosis of OSCC.

they found that there was no significant relation  between the risk of OSCC and any of the three specific SNPs. Nevertheless,  individuals with diabetes mellitus who possess a 22 minimum of one minor allele of rs10496915 (AC + CC; p = 0.046) were more inclined to develop large tumor 23 size in contrast to individuals who possess the main allele (AA) homozygously. 

the study is well-designed and written. However, the tables are not clear; an explanation regarding each parameter should be included in the table subtitles (for example, AA, AC ....). 

Reviewer 2 Report

Comments and Suggestions for Authors

Overall, the study is interesting. However, some parts are unclear and should be supplemented or corrected.

1. In the abstract is the sentence that conflicts with the results (lines 22-24)

2. Please add to the introduction information about studied SNPs (the place of mutation in genes and the consequences for their activity and expression). Why were these three SNPs chosen for investigation? NGS technology would allow for the investigation of many more SNPs. Why did you decide to use allelic discrimination?

3. The methodology 4.2 is too general. Please add information about the kit used in DNA extraction. Did you fraction blood samples before DNA isolation? Did you use quality and quantity verification of DNA before using it for PCR? What was the DNA amount used in PCR samples? The description of allelic discrimination using TaqMan probes should be more detailed. Please add sequences of primers and probes or their catalog number and manufacturer if you used commercial assays.

4. Did you try correlating the SNPs with an expression of LRPB1 in tumor cells?

5. The conclusions about the correlation between SNPs and patients suffering from OSCC and diabetes mellitus are based on too tiny groups. Thus, their worth is very limited. Please discuss it in the discussion. The last sentence in the conclusion about the possible use of rs6742944 in” predicting treatment response, drug toxicity, and recurrence in individuals with DM who have oral cancer” is exaggerating (lack of such studies in the literature)

6. I found a few editorial mistakes. The gene name was written in varying fonts: once in simple font and another in italics—please correct everywhere in italics. In line 20, the number of rs is wrong.
